# Development of a Portable, Ultra-Rapid and Ultra-Sensitive Cell-Based Biosensor for the Direct Detection of the SARS-CoV-2 S1 Spike Protein Antigen

**DOI:** 10.3390/s20113121

**Published:** 2020-05-31

**Authors:** Sophie Mavrikou, Georgia Moschopoulou, Vasileios Tsekouras, Spyridon Kintzios

**Affiliations:** Laboratory of Cell Technology, Department of Biotechnology, Agricultural University of Athens, EU-CONEXUS European University, 11855 Athens, Greece; sophie_mav@aua.gr (S.M.); geo_mos@aua.gr (G.M.); tsekouras@aua.gr (V.T.)

**Keywords:** Bioelectric Recognition Assay (BERA), membrane engineering, Point-of-Care (POC), S1 spike protein, serological assay, severe acute respiratory syndrome-coronavirus 2 (SARS-CoV-2)

## Abstract

One of the key challenges of the recent COVID-19 pandemic is the ability to accurately estimate the number of infected individuals, particularly asymptomatic and/or early-stage patients. We herewith report the proof-of-concept development of a biosensor able to detect the SARS-CoV-2 S1 spike protein expressed on the surface of the virus. The biosensor is based on membrane-engineered mammalian cells bearing the human chimeric spike S1 antibody. We demonstrate that the attachment of the protein to the membrane-bound antibodies resulted in a selective and considerable change in the cellular bioelectric properties measured by means of a Bioelectric Recognition Assay. The novel biosensor provided results in an ultra-rapid manner (3 min), with a detection limit of 1 fg/mL and a semi-linear range of response between 10 fg and 1 μg/mL. In addition, no cross-reactivity was observed against the SARS-CoV-2 nucleocapsid protein. Furthermore, the biosensor was configured as a ready-to-use platform, including a portable read-out device operated via smartphone/tablet. In this way, we demonstrate that the novel biosensor can be potentially applied for the mass screening of SARS-CoV-2 surface antigens without prior sample processing, therefore offering a possible solution for the timely monitoring and eventual control of the global coronavirus pandemic.

## 1. Introduction

The current outbreak of the severe acute respiratory syndrome-coronavirus 2 (SARS-CoV-2) has presented epidemiologists around the globe with an unpreceded challenge: the ability to reliably predict the spread of this novel highly contagious coronavirus and, in consequence, apply appropriate quarantine measures to prevent the transmission of the infection. In consequence, there is an urgent need for diagnostic tools able not only to reliably identify infected individuals—i.e., the source of infection—but also to determine if the infection is in the acute phase [1]. In addition, an indispensable goal for the control of the COVID-19 pandemic is the capacity for mass population screening, a condition that demands rapid and cost-efficient assay approaches. In response, a number of Point-of-Care (POC) rapid and relatively affordable tests for SARS-CoV-2 have been recently developed [2]. These are basically complementary to molecular tests, commonly operated in certified reference laboratories and mainly based on real-time reverse-transcriptase-based PCR (RT-PCR), with a limit of detection (LOD) of 4–10 copies/μL of the sample. Although molecular tests have the advantage of high sensitivity, they lack the high-throughput capacity required for mass population screening; for example, at least a couple of hours is required for the completion of the assay process, not including in this period the time required for sample collection, shipment and processing. Currently, a number of clustered regularly interspaced short palindromic repeats (CRISPR)/Cas-based rapid tests have been reported, the majority of them in the proof-of-concept stage of development, which target RNA sequences specific to SARS-CoV-2 [3]. These tests still require RNA extraction from the patient samples and further processing (e.g., the creation of ribonucleoprotein (RNP) complexes). On the other hand, serological tests targeting antibodies raised against viral envelope proteins are advantageous in terms of their lower cost and higher speed. However, they usually suffer from poor sensitivity. In addition, since the development of serum antibodies can take one to three weeks after SARS-CoV infection, the detection of antibodies in a patient sample does not reflect the viral load and/or the stage of virus replication in the host. Furthermore, host antibodies are usually detectable at least four days after infection.

Therefore, an immediate goal of the global management of the COVID-19 pandemic would be the ability to minimize the time required to confirm positive cases between infection and symptom appearance, preferably covering the very early infection period (1–3 days) and eventually allowing for monitoring the virus replication in asymptomatic patients. One of the most promising targets for the detection of SARS-CoV-2 are the viral surface spike proteins, which are the major immunodominant protein of SARS-CoV [4]. These are transmembrane glycoproteins responsible for receptor association, membrane fusion and viral entry. In particular, the S1 subunit is responsible for virus binding to cellular receptor(s), including angiotensin-converting enzyme 2 (ACE-2), which is the main target of the virus. S1 forms homotrimers protruding from the viral surface, thus mediating coronavirus entry to the host cells [5,6]. As is also known for SARS-CoV-2 and other pathogenic viruses, spike proteins can be used as reliable markers for the presence of infection and virus replication [7,8,9,10,11]. It has been also reported that S1 protein detection is more specific than other SARS-CoV structural proteins, such as the membrane (M), envelope (E) and nucleocapsid protein (NC) [12,13]. 

We herewith report the development of a novel biosensor for the ultra-rapid (3 min and sensitive -fg/mL level) detection of the SARS-CoV-2 S1 spike protein. Our method is based on mammalian Vero cells, which were engineered by electroinserting the human chimeric spike S1 antibody. This approach, known as Molecular Identification through Membrane Engineering, is a generic cell-based assay principle for the determination of analytes, including biomolecules, on the basis of the specific and selective interaction of the target analytes with cellular biorecognition elements, the surfaces of which have been modified by the electroinsertion of target-specific antibodies. It has been previously demonstrated that the binding of the target molecules to the electroinserted antibodies resulted in a unique and measurable change in the electric properties of the biorecognition elements, in particular the hyperpolarization of the engineered cell membrane [14,15,16]. These changes can be measured by any appropriate bioelectric/bioelectrochemical sensor. In the present study, we demonstrate that the binding of the SARS-CoV-2 S1 protein to the complementary antibody resulted in a considerable and selective change in the membrane-engineered cell bioelectric properties, which were measured by using a cell-biosensor set-up according to the principles of the Bioelectric Recognition Assay (BERA) [17,18]. Furthermore, we coupled the novel assay with a customized portable read-out device which was operated via smartphone/tablet. In this way, we also demonstrate the feasibility of applying the novel biosensor for the mass screening of SARS-CoV-2 surface antigens, offering the additional advantage of the direct monitoring of viral infection without prior sample processing. In this way, new perspectives are opened for managing the global coronavirus pandemic.

## 2. Materials and Methods

### 2.1. Green Monkey Kidney Cell Culture Conditions

The Vero cell line (ATCC CCL-81) was originally provided from the American Type Culture Collection (ATCC) (Manassas, VA, USA). The cells were cultured in Dulbecco’s Modified Eagle Medium (Biowest, Nuaillé, France) with the addition of 10% fetal bovine serum (Thermo Fisher Scientific, Waltham, MA, USA), 2 mM L-glutamine, 0.5 mM sodium pyruvate and 1% penicillin/ streptomycin (Biowest, Nuaillé, France) in an incubator (HF90Air jacketed CO_2_ Incubator, Heal Force Bio-Meditech Holdings Limited, Shanghai, China) at 5% CO_2_ and 37 °C. Subcultures were performed twice per week after detachment with the use of trypsin/ethylenediaminetetraacetic acid (EDTA) for 10 min at 37 °C.

### 2.2. Sensor Fabrication from Membrane-Engineered Vero Cells (Vero/Anti-S1)

Membrane-engineered kidney cells were created by the electroinsertion of the SARS-CoV-2 Spike S1 antibody (HC2001, GenScript Biotech, Leiden, Netherlands) into the cell membrane according to previously described protocols [19,20]. Briefly, theVero cells were detached and collected after centrifugation (4 min, 1000 rpm, 25 °C), with a final density of 2.5 × 10^6^/mL. The cell pellet was resuspended in phosphate-buffered saline (PBS) (pH 7.4) containing various antibody concentrations (0.5, 1, 5 and 10 μg/mL) and incubated for 20 min at 4 °C. Afterwards, the cell suspension was transferred to electroporator (Eppendorf Eporator, Eppendorf AG, Hamburg, Germany) cuvettes. The electroinsertion was followed by the application of two pulses of an electric field at 1800 V/cm. Then, the cells were transferred to a petri dish (60 × 15 mm) containing cell culture medium and maintained in the incubator overnight. The next day, the Vero/anti-S1 cells were mechanically detached from the dish and collected with PBS (pH 7.4) in Eppendorf tubes. Based on the calculation by Zeira et al. [21] and under the experimental conditions applied in the present study (0.5 μg/mL antibody per 2.5 × 10^6^ cells), we estimate that a maximum quantity of 80 fg anti-S1 antibody (approx. 12 × 10^4^ molecules) will be inserted per cell. In addition, Kokla et al. [16] showed that the cell membrane can be essentially saturated with electroinserted antibodies in membrane-engineered cells.

### 2.3. Vero/Anti-S1 Cell Membrane Potential Measurements: Biosensor Set-Up

As previously reported [16,19], the membrane potential of membrane-engineered Vero cells is affected by the interactions of electroinserted receptor molecules and the analyte anions, producing measurable changes in the membrane potential. A customized multichannel potentiometer (EMBIO DIAGNOSTICS Ltd., Strovolos, Cyprus) was used to record the Vero/anti-S1 cell membrane potential changes according to the principle of the Bioelectric Recognition Assay (BERA) [17,18]. The device measures the actual availability (binding and/or uptake) of analytes through changes in the membrane potential and other electric properties of the cells in a highly sensitive, rapid, high-throughput and reproducible mode. The device is capable of measuring real-time changes in the cells’ electric properties (up to eight simultaneous measurements) from eight gold screen-printed electrodes (working electrode: gold (Au), reference: Ag/AgCl) on a disposable sensor strip (iMiCROQ S.L., Tarragona, Spain) (Figure 1a).

In order to avoid sample diffusion between the different screen printed electrodes on the same strip, a custom fabricated polydimethylsiloxane (PDMS) layer with eight wells (diameter = 4 mm) was attached by the use of a cyanoacrylate adhesive (Fevikwik, PIDILITE, Andheri, India) on the electrode’s polyester part, containing eight holes corresponding to the respective positions of the screen printed electrodes. Initially, the PDMS base and curing reagent (SYLGARD^®^ 184 Silicone Elastomer Kit, Sigma-Aldrich, Saint Louis, MO, USA) were mixed thoroughly and poured into a 3D-printed polylactic acid (PLA) mold. The filled mold was baked at 60 °C for 120 min and the cured PDMS substrate was removed from the mold and baked once again at 110 °C.

The PLA mold used for the PDMS layer fabrication was designed with the 123D Design software (Autodesk, San Rafael, CA, USA). A CelRobox 3D printer device (RoboxDual, CEL-UK, Bristol, UK) was used for the mold printing by the application of the fused deposition modeling technique (FDM), where the filament that passes through the heated print head lays down on a construction platform in a layer-by-layer fashion until the object’s form is complete [22]. The nozzle diameter of the print head was 0.4 mm and the printing temperature for the PLA was 200 °C. In addition, the PLA filament’s diameter was 1.75 ± 0.05 mm (CEL-UK, Bristol, UK). After the printing process, the wells were sterilized with 70% (*v/v*) ethanol for 10 min then dried for 2 h under a sterile hood.

Membrane-engineered Vero cells in suspension were added first to the wells of the PDMS layer at the top of each of the eight gold screen-printed electrodes contained in each disposable sensor strip (20 μL ≈ 5 × 10^4^ cells) with the help of a multichannel automatic pipette (Figure 1b). Then, 20 μL of the sample (either standard solution of the SARS-CoV-2 Spike protein (S1) or SARS-CoV-2 nucleocapsid protein) (GenScript Biotech B.V., Netherlands) was added, and the response of the cells was immediately recorded as a time-series of potentiometric measurements (in Volts) through custom-made software (B.EL.D., EMBIO DIAGNOSTICS Ltd., Cyprus) (Figure 1c). Each measurement lasted 10 min and 1200 values/sample were recorded (Figure 1d).

### 2.4. Viability Monitoring of Membrane-Engineered Cells

Cell morphology changes were captured 0, 24 and 48 h after various concentrations of the SARS-CoV-2 Spike S1 antibody (0, 0.5, 1, 5 and 10 μg/mL) electroporation by an inverted microscope (ZEISS Axio Vert.A1, Carl Zeiss Microscopy, LLC, White Plains, NY, USA), and the pictures were processed by ZEN lite software.

### 2.5. Data Analysis and Experimental Design

The present study was implemented in three experimental sections: the first experiment concerned the investigation of the effect of the concentration (0.5, 1, 5 or 10 μg/mL) of the electroinserted antibody on the response of membrane-engineered Vero/anti-S1cells to 5 μg/mL of the SARS-CoV-2 spike S1 protein. This relatively high S1 protein concentration was chosen in order to guarantee a sufficiently high number of antibody-antigen binding interactions. In the second experiment, the biosensor response against a series of S1 protein concentrations was investigated. A range of 1 fg-2.5 μg/mL was chosen to reflect the usual concentrations of viral protein in vivo. Finally, the third experiment was designed to allow for assessing the cross-reactivity of the sensor against the SARS-CoV-2 nucleocapsid protein.

The response of Vero/anti-S1 cells immediately after the addition of the sample was recorded as a time-series of potentiometric signal (measured in Volts) at a sampling rate of 2 Hz. The measurements were uploaded via a tablet/Bluetooth communication to a cloud server [23]. The experimental set-up was structured in a completely randomized design with three technical replicates, while a set of eight biosensors was tested against every single sample—i.e., the response of each sample x biorecognition element combination was measured eight times by independent assays, while each experimental set was repeated at three different dates (n = 24). All the results are expressed as the mean ± SEM. Differences between the means were tested for statistical significance using an analysis of variance. 

## 3. Results

### 3.1. Membrane-Engineered Vero/anti-S1 Cells have a Distinct Response Against the SARS-CoV-2 Spike S1 Protein

The membrane engineering procedure using different antibody concentrations did not affect the cell morphology and adherence 24 and 48 h after application, as can be seen in Figure 2.

Upon exposure to 5 μg/mL of the SARS-CoV-2 spike S1 protein, the Vero/anti-S1 cells which were membrane-engineered with the respective human chimeric antibodies demonstrated a strong response expressed by a considerable reduction in the biosensor’s potential. This response was very rapid (measurable after three minutes of protein–cell interaction) and clearly distinguishable from the response of non-electroporated Vero cells or electroporated but not antibody-engineered cells (Figure 3) (p < 0.0001). Additionally, the response of the membrane-engineered cells was entirely distinct from the response recorded upon the addition of the S1 protein solution to cell-free electrodes. In addition, statistically significant differences were observed among the responses of Vero cells, which were membrane-engineered with different antibody concentrations (0.5–10 μg/mL).

### 3.2. The Biosensor Response is Dependent on the Concentration of the SARS-CoV-2 Spike S1 Protein

A concentration-dependent response was observed during the assay of increasing concentrations of the SARS-CoV-2 spike S1 protein with the novel biosensor based on Vero/anti-S1 cells membrane-engineered with 0.5 μg/mL human chimeric anti-S1 antibodies (Figure 4), with a linear pattern in the range 10 fg–1 μg/mL. This corresponds to 15 × 10^2^–15 × 10^11^ S1 protein molecules per sample per cell population (50 × 10^3^ cells) (a sample volume of 20 μL is applied in the engineered cell population). Measurements at each S1 protein concentration were distinct and significantly different from the control solutions (i.e., zero S1 concentration). Results were quite reproducible, with an average variation of 8.7% over all assayed concentrations and a limit of detection (LOD) of one (1) fg/mL, which was significantly greater than the 3× standard deviation of the response against the control.

### 3.3. The Biosensor Response is Selective for the SARS-CoV-2 Spike S1 Protein

The biosensor based on Vero/anti-S1 cells membrane-engineered with 0.5 μg/mL of human chimeric anti-S1 antibodies was also tested against the SARS-CoV-2 nucleocapsid (NC) protein (1 fg/mL–100 pg/mL) (Figure 5). In every tested NC concentration, the observed response was either lower than or equal to the control (zero concentration)—i.e., it was entirely distinct and opposite from the response against similar concentrations of the spike S1 protein, which was higher than the control (comp. Figure 4). In other words, as far as these two SARS-CoV-2 proteins were concerned, a selective assay was determined for the S1 protein.

## 4. Discussion

The infection potential of SARS-CoV-2 is particularly high due to its long transmission period [24], meaning that asymptomatic COVID-19 patients are able to transmit the virus during their incubation periods (2–14 days) [25]. Currently available serological assays for SARS-CoV-2 screen host antibodies raised against the virus, targeting different immunoglobulin (Ig) subclasses—mainly Immunoglobulin M (IgM) (usually earlier appearing but with lower antigen affinity) and Immunoglobulin G (IgG)—therefore allowing for a comparative discrimination between patients in an early infection stage (IgM, 4–10 days) or a late one (IgG, 11 days or later) [26]. Serological tests do not allow for the detection of infection at an earlier stage (0–3 days); however, this is feasible if viral antigens are screened instead of host antibodies. In particular, the S1 spike protein of SARS-CoV-2 is a key contributor to the early stages of infection, since it facilitates the initial stages of viral entry and serves as the major target recognized by humoral and cellular immune responses [27,28], therefore its detection corresponds to the presence of the whole virus. S proteins are key elements in discriminating between different coronaviruses; for example, the SARS-CoV-2 spike protein binds to human ACE2 with an almost twentyfold higher affinity than the respective SARS-CoV spike protein [29].

There is currently limited progress in the sensitive detection method for the SARS-CoV S1 protein. [30] reported a sandwich ELISA assay employing monoclonal and bi-specific monoclonal anti-S1 antibodies, with a detection limit of 0.019 μg/mL when using bi-specific monoclonal antibodies as the detection antibodies. According to the assay protocol, an incubation stage at 37 °C for 1 h was required, while the development of the final optical (colorimetric) signal at 650 nm required another 15 min. The assay could provide quantitative determination in a range of 0.0047-1.25 μg/mL S1 protein concentration.

Very recently, Seo et al. [31] reported a field-effect transistor (FET)-based biosensing device in which the sensing area was the SARS-CoV-2 spike antibody immobilized onto a graphene sheet by 1-pyrenebutyric acid N-hydroxysuccinimide ester (PBASE) probe linker. The sensor was able to detect the S1 spike protein with high sensitivity in phosphate-buffered saline as well as clinical samples. However, the measuring set-up used required the use of costly and low-throughput instrumentation (semiconductor analyzer), as well as considerably high antibody concentrations (250 μg/mL) for the functionalization of the fabricated graphene-based device. In addition, a dual-functional plasmonic biosensor integrating the plasmonic photothermal (PPT) effect and localized surface plasmon resonance (LSPR)-sensing transduction on a chip containing gold nanoabsorbers (AuNIs) has been recently reported as an alternative solution for the detection of COVID-19 viral sequences [32]. The LSPR biosensor demonstrated a considerable sensitivity towards SARS-CoV-2 sequences, with a detection limit down to the concentration of 0.22 pM.

In the present study, we applied a methodological approach known as Molecular Identification through Membrane Engineering. This is a generic methodology of artificially inserting tens of thousands of antibody molecules on the cell surface, thus rendering the cell a selective responder against antigens binding to the inserted antibodies. The working assumption of the method is that the attachment of the target antigen to its respective antibody causes a change in the cell membrane structure, which is measurable as a change in the cell membrane potential. Kokla et al. [16] used purified anti-biotin antibodies from rabbit antiserum along with in-house-prepared biotinylated bovine serum albumin (BSA) as a model antibody–antigen pair of molecules for facilitating membrane engineering experiments. They have proven that (i) membrane-engineered cells incorporated the specific antibodies in the correct orientation, and that (ii) the inserted antibodies selectively interact with the homologous target molecules. Furthermore, membrane-engineered cells have been used as biorecognition elements in bioelectric sensors based on the principle of the Bioelectric Recognition Assay [17,18] and successfully applied for the detection of various human and plant viruses [19,33,34,35]. Similar to the results of the present study, the majority of these previous publications reported the hyperpolarization of the engineered cell membrane as a result of the interaction of the antibody-bearing membrane-engineered cells with the antigens under determination. This was demonstrated by the decrease in the normalized biosensor response with increasing concentrations of the SARS-CoV-2 spike S1 protein in the linear range of response (10 fg/mL–1μg/mL). It should be mentioned that an opposite pattern (increasing biosensor response) was observed in the lowest concentration range between 0 and 10 fg/mL. This can be explained by the following hypothesis: 10 fg/mL of S1 protein corresponds approximately to 1500 viral protein molecules per 50,000 membrane-engineered cells, which are present in each assay. At the same time, when 0.5 μg/mL of antibody is used, each cell bears 12 × 10^4^ electroinserted antibodies, meaning that the cell population used in each assay presents 6 × 10^9^ antibody molecules to the sample. In other words, the antibody:protein ratio at 1, 5 and 10 fg/mL of S1 protein is, respectively, 0.4 × 10^6^, 2 × 10^6^ and 4 × 10^6^. This ratio is sufficient to guarantee that the majority of the S1 protein will bind to the membrane-inserted antibodies. However, as the added S1 concentrations increase (above 10 fg/mL), two conflicting mechanisms could take place: on one hand, an increased number of S1 molecules would lead to an increased number of protein–antibody interactions; on the other hand, increased numbers of protein molecules would compete with each other for antibody-binding positions on the engineered cell membrane, therefore reducing the availability of membrane-bound antibodies (for example, the antibody:protein ratio becomes 0.4 × 10^−3^ at 1 μg/mL, i.e., 10^9^ times lower than at 1 fg/mL). This would possibly result in partially reduced changes in the cell membrane potential. This pattern of response is frequently observed in membrane-engineered cells.

Using the novel biosensor, we were able to discriminate different S1 protein concentrations (including control) within just three minutes of total assay duration. That said, the extension of the assay time to ten minutes improved the resolution of the response between different concentrations.

Application-wise, the novel biosensor was developed on the principle of providing a truly portable, high throughput and low-cost system for the mass screening of the coronavirus antigens. Indeed, the biosensor set-up presented in this study allows for the mass manufacturing of the customized low-cost potentiometer, while thousands of test consumables (membrane-engineered cells) can be prepared using just 0.5 μg/mL of anti-S1 antibody. In addition, the observed high sensitivity of the biosensor could allow for screening the virus in easy-to-obtain patient samples such as saliva [36].

By using antibodies, enzymes (e.g., ACE2) or other receptor-like molecules (e.g., aptamers) against distinct domains within the S1 subunit, the novel biosensor assay could also be used to detect different coronaviruses [6,37,38,39,40]. Naturally, other factors should be additionally considered in such cases—for instance, the co-insertion of proteases for the facilitation of protein structural changes necessary for receptor engagement [41].

In spite of the fact that membrane engineering has been methodologically standardized and tested in several applications during the last 15 years, it should be expected that cell viability and membrane-engineering efficiency will differ, even slightly, between different batches of engineered cells, not at least due to the different passages of cultured cells. For this reason, various sets of controls (non-electoporated cells, electroporated but non-engineered cells) must be included in each individual measurement within every batch of cells.

Cell-based biosensors present unique challenges for mass-scale, regular on-site applications. The most critical issue is reduced cell viability. Various approaches have been proposed to overcome this limitation, including the use of microfluidic/organ-on-chip circuits in biosensing platforms or specific cell types (e.g., fish gills). However, these approaches have not been yet proven to be suitable for the practical cost-efficient operation of cell-based biosensors for routine mass-screening applications.

Based on our past experience with the commercial applications of membrane-engineered cells and according to the results of the present study, we know that cell viability and biosensor response remain unaltered at least three days after electroinsertion. Cells prepared on day 1 could be shipped and used on-site during the next two days. As it stands, this set-up allows for using the biosensor by relatively locally situated end-users, preferably availing over controlled environments (e.g., diagnostic laboratories). We have calculated the actual logistics of the capacity of a cell culture laboratory to produce enough biorecognition elements to meet the local (e.g., within a city or small region) screening requirements. At least 45x10^6^ membrane-engineered cells could be manufactured by a skilled technician on a daily basis, corresponding to more than 100 tests (including negative and positive controls). We have previously reported [42] that the immobilization of membrane-engineered cells in calcium alginate gels and their consequent storage in medium supplemented or not with 20% (*v/v*) fetal calf serum enabled the preservation of at least 90% of viable cells four weeks after electroinsertion. The cell proliferation could be controlled by adjusting the concentration of fetal calf serum. This has allowed the practical shipment and use of immobilized membrane-engineered cells by remote end-users in a number of applications. Cell immobilization in gel has not been used in the present report; we definitely plan to investigate this possibility in the next round of biosensor development and optimization.

## 5. Conclusions

The present report is basically the proof of the methodological concept of the novel biosensor assay for the detection of the SARS CoV-2 spike S1 protein. The next step will be the actual clinical validation of the assay using patient samples and comparison to current serological and molecular tests. In parallel, we plan to optimize the assay by expanding the number of cell lines to be membrane-engineered with the human chimeric spike S1 antibody and by further investigating the cross-reactivity and specificity of the biosensor, in particular against the S proteins of other coronaviruses. Finally, we are currently working on improving the interface of the read-out device with an embedded software able to present to the end user with final results as a functional decision-support tool. In this way, we hope to be able to provide an efficient monitoring tool to assist transmission pattern analysis and the identification of asymptomatic cases as a new contribution to the global effort to manage the coronavirus pandemic.

## Figures and Tables

**Figure 1 sensors-20-03121-f001:**
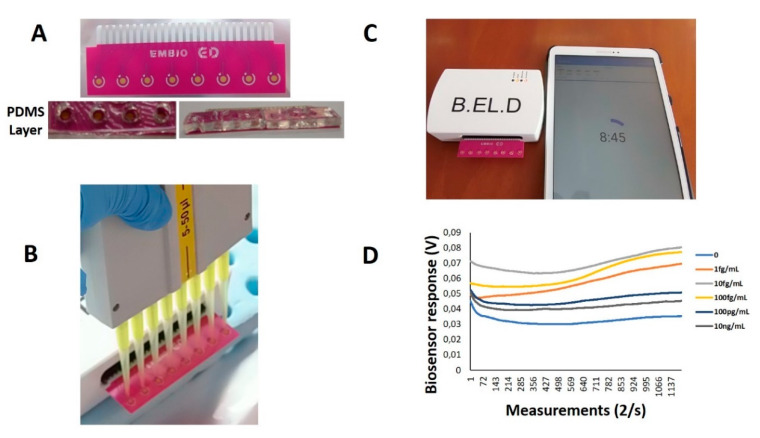
Experimental set-up of the Vero/anti-S1 cell-based biosensor’s assembly. An eight-channel gold screen-printed electrode assembly was prepared with the PDMS layer attached for the well formation (**A**). The potentiometer device is connected to a tablet device for the recording of the measurements immediately after the sample application (**B**,**C**). The electric signal is visualized through a voltage vs. time graph (**D**).

**Figure 2 sensors-20-03121-f002:**
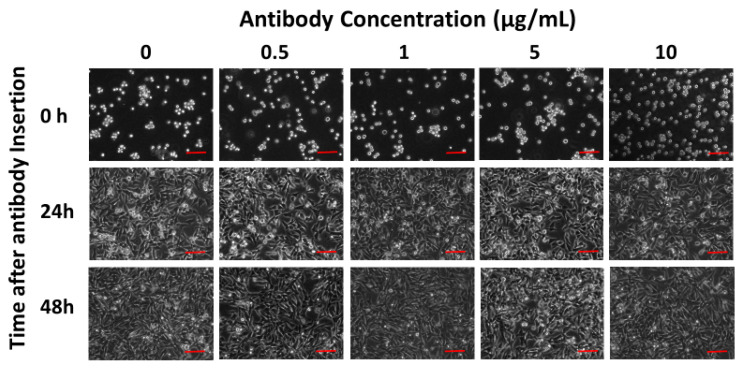
Morphological changes in the Vero cells 0, 24 and 48 h after electroinsertion of 0, 0.5, 1, 5 and 10 μg/mL of SARS-CoV-2 Spike S1 antibody. Scale bars = 50 μm.

**Figure 3 sensors-20-03121-f003:**
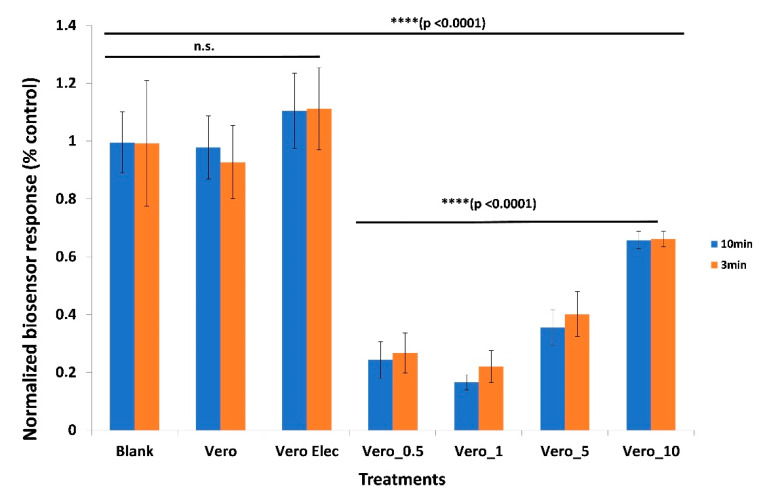
Distinct response of Vero/anti-S1 cells membrane-engineered with human chimeric antibodies (Vero_) against the SARS-CoV-2 spike S1 protein compared with the response of non-electroporated Vero cells (Vero) or electroporated but not antibody-engineered cells (Vero Elec). Cells were membrane-engineered with different antibody concentrations (Vero_0.5:0.5 μg/mL; Vero_1:1 μg/mL; Vero_5:5 μg/mL; Vero_10:10 μg/mL). Blank: response of the S1 protein solution added to cell-free electrodes. Results are presented after three (red columns) or ten minutes (blue columns) of sample–cell interaction. ****: statistically significant different results (p < 0.0001); n.s.: non-statistically significant different results. Results are expressed as normalized biosensor responses (% blank = control) (n = 24).

**Figure 4 sensors-20-03121-f004:**
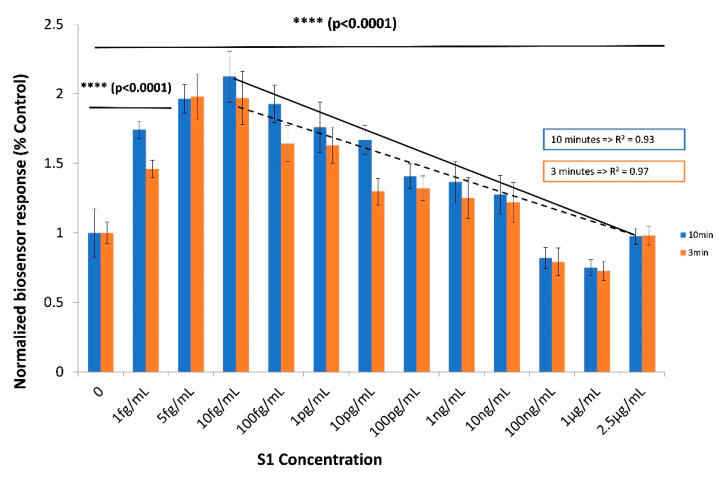
Concentration-dependent biosensor responses against the SARS-CoV-2 spike S1 protein. Vero/anti-S1 cells membrane-engineered with 0.5 μg/mL of human chimeric antibodies were used as the biorecognition element. Results are presented after three (red columns) or ten minutes (blue columns) of sample–cell interaction. ****: statistically significant different results (p < 0.0001). Results are expressed as normalized biosensor responses (% control) (n = 24).

**Figure 5 sensors-20-03121-f005:**
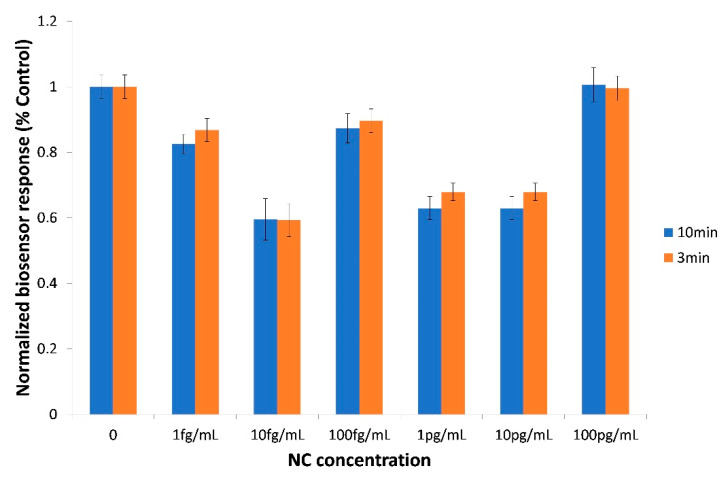
Biosensor cross-response against the SARS-CoV-2 nucleocapsid (NC) protein in the 1fg/mL–100 pg/mL concentration range. Vero/anti-S1 cells membrane-engineered with 0.5 μg/mL of human chimeric antibodies were used as the biorecognition element. Results are presented after three (red columns) or ten minutes (blue columns) of sample–cell interaction. Results are expressed as normalized biosensor responses (% control) (n = 24).

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
