# Peer review of "Development of a Portable, Ultra-Rapid and Ultra-Sensitive Cell-Based Biosensor for the Direct Detection of the SARS-CoV-2 S1 Spike Protein Antigen"

_sensors, 2020, doi:10.3390/s20113121_

Round 1
Reviewer 1 Report
This manuscript by Mavrikou et al. described a potentiometric method for S1 spike protein antigen detection. The idea is interesting, but some additional results need be presented to support the conclusion. Following are some suggestion,
- Please give the real time detection curves upon the detection of spike protein antigen.
- The working performance of the fabricated need be carried out in detail, such as the linear range, sensitivity et al.
Author Response
Comment 1: Please give the real time detection curves upon the detection of spike protein antigen.
Action taken: Figure 1D was replaced by one presenting actual series of the biosensor’s response against increasing concentrations of the S1 spike protein.
Comment 2: The working performance of the fabricated need be carried out in detail, such as the linear range, sensitivity et al.
Response: The data presented in the submitted manuscript resulted both from replications of each experiment as well as experiments conducted on different dates. Additional experiments were carried out following the submission of the manuscript in order to possibly extend the range of linear response. In this respect, we were able to determine a linear range of response beyond 100 pg/mL and up to 1μg/mL. In addition, the established limit of detection was confirmed, the biosensor’s response at the LOD being significantly greater than 3 times the standard deviation of the response against control.
Action taken:
- The following was added in 2.5. Data analysis and experimental design (lines 171-173 of the revised manuscript):
…, i.e. the response of each sample x biorecognition element combination was measured eight times by independent assays, while each experimental set was repeated at three different dates (n=24).
- The number of replications (n= 24) was also added in the end of the Figure legends.
- The range of linear response was determined to be 10 fg/mL – 1 μg/mL. This was respectively reported in subsection 3.2 of the revised manuscript (lines 208-210).
- The following was added in the end of 3.2.:
…, which was significantly greater than 3 x standard deviation of the response against control.

Reviewer 2 Report
This is a very interesting study presenting a novel way of detecting viral proteins - as the authors claim. The study is well written, but in my opinion the work must be significantly improved to convince the reader that their actual results are valid.
- Much more details of the membrane engineering should be given. How many Ab is inserted into one single cell etc.
- Pictures of cells, viability of engineered cells would be useful to show.
- A simple calculation must support the data. Ab/cell and viral protein/cell in a typical sample.
- In my opinion the results in Fig. 3 are strongly questionable. Why the response to 1 fg/mL is around the response of 1-10 pg/mL. Why the response is largest for the 10 fg/mL sample? I do not really see a linear response here. Also, much broader concentration ranges must be investigated. Why to show only the responses to such a low concentration levels? The reader is very much interested to see the saturation of this sensor response by increasing sample concentrations. The responses for ng/mL, microg/mL range should be checked, too.
- Time dependent signals after sample addition should be shown, too.
- Also, the above result must be shown - in the whole concentration range - for control proteins not binding to the Abs.
- When claiming such a low sensitivities, blank injections (buffer without sample) should be incorporated between sample injections, and the signal changes must be shown in time. Also, non-binding protein injection must be incorporated in control experiments.
Overall, Fig. 3 did not convince me that these results are valid and the sensor is measuring such a low concentration levels, or it is actually measuring the presence of viral proteins...
Author Response
The comments were quite constructive and we have taken them in thoughtful consideration. The authors wish to thank the reviewers for helping improve our manuscript. In following, we respond to each point raised by each Reviewer, along with associated changes in the revised manuscript.
Comment 1: Much more details of the membrane engineering should be given. How many Ab is inserted into one single cell etc.
Response: Based on the calculation by Zeira et al. [21] and under the experimental conditions applied in the present study (0,5 μg/mL antibody per 2,5 × 106 cells) we estimate that a maximum quantity of 80 fg anti-S1 antibody (approx. 12 x 104 molecules) will be inserted per cell. In addition, Kokla et al. [16] showed that the cell membrane can be essentially saturated with electroinserted antibodies in membrane-engineered cells.
Action taken: The following statement was added in the Materials and Methods (subsection 2.2.) of the revised manuscript (lines 106-110):
Based on the calculation by Zeira et al. [21] and under the experimental conditions applied in the present study (0,5 μg/mL antibody per 2,5 × 106 cells) we estimate that a maximum quantity of 80 fg anti-S1 antibody (approx. 12 x 104 molecules) will be inserted per cell. In addition, Kokla et al. [16] showed that the cell membrane can be essentially saturated with electroinserted antibodies in membrane-engineered cells.
Additional reference:
[21] Zeira, M.; Tosi, P.F.; Mouneimne, Y.; Lazarte, J.; Sneed, L.; Volsky, D.L.; Nikolau, C. Full-length CD4 electroinserted in the erythrocyte membrane as a long-lived inhibitor of infection by human immunodeficiency virus. Proc. Natl. Acad. Sci. U.S.A. 1991, 88, 4409-4413. DOI: 10.1073/pnas.88.10.4409
Comment 2: Pictures of cells, viability of engineered cells would be useful to show.
Response: Based on our past experience with commercial applications of membrane-engineered cells, we know that cell viability and biosensor response remain unaltered at least three days after electroinsertion. Cells prepared on day 1 could be shipped and used on-site during the next two days. We have previously reported [42] that the immobilization of membrane-engineered cells in calcium alginate gels and their consequent storage in medium supplemented or not with 20% (v/v) foetal calf serum enabled the preservation of at least 90% of viable cells four weeks after electroinsertion. Cell proliferation could be controlled by adjusting the concentration of foetal calf serum. Cell immobilization in gel has not been used in the present report; we definitely plan to investigate this possibility in the next round of the biosensor development and optimization (see also response to Comment #1, Reviewer #3).
Regarding the present study and in response to the Reviewer’s comment, we have conducted a comparative viability test for control and membrane-engineered Vero cells on 24 ad 48 hrs after electroinsertion. The result of the test demonstrated that the membrane engineering procedure using different antibody concentrations did not affect cell morphology and adherence 24 and 48h after application. Pictures of viable membrane-engineered cells can also be found in [14-16, 19, 32, 33].
Action taken:
- Subsection 2.4 was added in the Materials and Methods section of the revised manuscript (lines 152-156) describing the methodology for the viability assays.
- A new figure (Figure 2) was added presenting the results of the viability tests. The following figures were renumbered.
- The following statement was added in the beginning of subsection 3.1.:
The membrane engineering procedure using different antibody concentrations did not affect cell morphology and adherence 24 and 48h after application, as can be seen in Figure 2.
- The following statement was added in the end of the Discussion of the revised manuscript (lines 327-343):
Based on our past experience with commercial applications of membrane-engineered cells and according to the results of the present study, we know that cell viability and biosensor response remain unaltered at least three days after electroinsertion. Cells prepared on day 1 could be shipped and used on-site during the next two days. As it stands, this set up allows for using the biosensor by relatively locally situated end-users, preferably availing over controlled environments (e.g. diagnostic laboratories). We have calculated the actual logistics of the capacity of a cell culture laboratory to produce enough biorecognition elements to meet local screening requirements, e.g. within a city or small region. At least 45 million membrane-engineered cells could be manufactured by a skilled technician on a daily basis, corresponding to approx. 100 tests/day. We have previously reported [42] that the immobilization of membrane-engineered cells in calcium alginate gels and their consequent storage in medium supplemented or not with 20% (v/v) foetal calf serum enabled the preservation of at least 90% of viable cells four weeks after electroinsertion. Cell proliferation could be controlled by adjusting the concentration of foetal calf serum. This has allowed the practical shipment and use of immobilized membrane-engineered cells by remote end-users in a number of applications. Cell immobilization in gel has not been used in the present report; we definitely plan to investigate this possibility in the next round of the biosensor development and optimization. (see also response to Comment #1, Reviewer #3)
Additional reference:
[42] Katsanakis N,; Katsivelis, A; Kintzios, S. Immobilization of Electroporated Cells for Fabrication of Cellular Biosensors: Physiological Effects of the Shape of Calcium Alginate Matrices and Foetal Calf Serum. Sensors 2009, 9, 378-385. DOI: 10.3390/s90100378
Comment 3: A simple calculation must support the data. Ab/cell and viral protein/cell in a typical sample.
Response: As mentioned in response to Comment #1 above, a maximum quantity of 80 fg of antibody/cell is electroinserted when 0,5 μg/mL of antibody solution is used. This corresponds to 12 x 104 antibody molecules/cell. As far as the S1 concentration is concerned, the range of tested concentrations (1 fg/mL -2.5 μg/mL ) corresponds to 154 – 15x1010 S1 protein molecules per sample per cell population (50 x 103 cells) ( a sample volume of 20 μL is applied in the engineered cell population).
Action: Corresponding S1 protein concentrations/cell were added in subsection 3.2. of the revised manuscript (lines 208-210).
Comment 4: In my opinion the results in Fig. 3 are strongly questionable. Why the response to 1 fg/mL is around the response of 1-10 pg/mL. Why the response is largest for the 10 fg/mL sample? I do not really see a linear response here. Also, much broader concentration ranges must be investigated. Why to show only the responses to such a low concentration levels? The reader is very much interested to see the saturation of this sensor response by increasing sample concentrations. The responses for ng/mL, microg/mL range should be checked, too.
Response: As explained above (response to Comments #1 and #3), 10 fg/mL S1 protein correspond approximately to 1500 viral protein molecules per 50,000 membrane-engineered cells, which are present in each assay. At the same time, when 0.5 μg/mL of antibody is used, each cell bears 12 x 104 electroinserted antibodies, meaning that the cell population used in each assay presents 6 x 109 antibody molecules to the sample. In other words, the antibody:protein ratio at 1, 5 and 10 fg/mL S1 protein is, respectively,0.4x106, 2x106 and 4x106. This ratio is sufficient to guarantee that the majority of S1 protein will bind to the membrane-inserted antibodies. However, as added S1 concentrations increase (above 10 fg/mL), two conflicting mechanisms could take place: on one hand, an increased number of S1 molecules would lead to increased number of protein-antibody interactions; on the other hand, increased numbers of protein molecules would compete with each other for antibody-binding positions on the engineered cell membrane, therefore reducing the availability of membrane-bound antibodies (for example, antibody:protein ratio becomes 0.4x10-3 at 1 μg/mL, i.e. 109 times lower than at 1 fg/mL). This would possibly result in partially reduced changes of the cell membrane potential. This pattern of response is frequently observed in membrane-engineered cells.
Action taken:
The following statement was added in the Discussion of the revised manuscript (lines 285-300):
It should be mentioned that an opposite pattern (increasing biosensor response) was observed in the lowest concentration range between 0 and 10 fg/mL. This can be explained by the following hypothesis: 10 fg/mL S1 protein correspond approximately to 1500 viral protein molecules per 50,000 membrane-engineered cells, which are present in each assay. At the same time, when 0.5 μg/mL of antibody is used, each cell bears 12 x 104 electroinserted antibodies, meaning that the cell population used in each assay presents 6 x 109 antibody molecules to the sample. In other words, the antibody:protein ratio at 1, 5 and 10 fg/mL S1 protein is, respectively,0.4x106, 2x106 and 4x106. This ratio is sufficient to guarantee that the majority of S1 protein will bind to the membrane-inserted antibodies. However, as added S1 concentrations increase (above 10 fg/mL), two conflicting mechanisms could take place: on one hand, an increased number of S1 molecules would lead to increased number of protein-antibody interactions; on the other hand, increased numbers of protein molecules would compete with each other for antibody-binding positions on the engineered cell membrane, therefore reducing the availability of membrane-bound antibodies (for example, antibody:protein ratio becomes 0.4x10-3 at 1 μg/mL, i.e. 109 times lower than at 1 fg/mL). This would possibly result in partially reduced changes of the cell membrane potential. This pattern of response is frequently observed in membrane-engineered cells.
Comment 5: Time dependent signals after sample addition should be shown, too.
Action taken: Figure 1D was replaced by one showing presenting actual series of the biosensor’s response against increasing concentrations of the S1 spike protein.
Comment 6: Also, the above result must be shown - in the whole concentration range - for control proteins not binding to the Abs.
Response: Following additional experimentation, Figure 5 (formerly Figure 4) was expanded to demonstrate the selective biosensor response against the SARS-CoV-2 spike S1 protein compared to the SARS-CoV-2 nucleocapsid (NC) protein in the 1 fg/mL- 100 pg/mL concentration range.
Action taken: Figure 5 (formerly Figure 4) was redrawn along with its legend. Subsection 3.3. was rewritten as follows:
The biosensor, based on Vero/anti-S1 cells membrane-engineered with 0.5 μg/mL human chimeric anti-S1 antibodies, was also tested against the SARS-CoV-2 nucleocapsid (NC) protein (1 fg/mL - 100 pg/mL) (Figure 5). In every tested NC concentration, the observed response was either lower or equal to the control (zero concentration), i.e. it was entirely distinct and opposite from the response against the similar concentrations of the spike S1 protein, which was higher from the control (comp. Figure 4). In other words, as far these two SARS-CoV-2 proteins were concerned, a selective assay was determined for the S1 protein.
Comment 7: When claiming such a low sensitivities, blank injections (buffer without sample) should be incorporated between sample injections, and the signal changes must be shown in time. Also, non-binding protein injection must be incorporated in control experiments.
Response: In between sample (blank or loaded) injections are not compatible with our method, since we consider that the cellular biorecognition elements may not return to a steady-state condition after interaction with the S1 protein. Contrary to conventional serology/antigen assays, no washing steps are foreseen in our procedure; such would also impose stress on the cells, thus rendering unreliable the interpretation of the results.

Reviewer 3 Report
The authors present a demonstration of concept for a high sensitivity viral protein detection, with the development of the accompanying set of accessories to be used a high throughput field screen test.
This original approach is conceptually intriguing but bears practical challenges in field implementation. The limiting factor of this approach seems to be the preparation of cell culture with the membrane inserted antibody, as the culture to be used for the diagnostic test will need to be prepared de novo for each batch, thus presuming tissue culture lab, reasonably adjacent to the test site. The use of live transiently modified cells also raises several questions as to the feasibility of the high throughput test that should be addressed in the manuscripts: what would be the setup and practical scale for producing these cells, how often the cells need to be regenerated (practically), how long they can stay in culture to preserve they sensing capabilities, can they be frozen, how would time after the modification affects the sensitivity of the cells? This should be discussed in the context of a precedent (or the lack of it).
Establishing the variability between iterations of membrane engineering step is critical as this step needs to be repeated for every batch. This variability of the transient cell modification will also emanate from dilution of the probe as cells divide.
Fig. 2 Vero10 is significantly higher than Vero5, as plateau was not reached, would Vero15 be better?
Though qualitatively the response of the modified cells to the antigen exposure is clear, the estimation of variance and reproducibility are key for a tool aiming at mass diagnostics. Therefore, statistics calculations should be presented in greater detail, including the test for significance and the variance estimate (error bar description, what is the nature and number of repeats and statistic parameter they represent, in the legends).
Author Response
The comments were quite constructive and we have taken them in thoughtful consideration. The authors wish to thank the reviewers for helping improve our manuscript. In following, we respond to each point raised by each Reviewer, along with associated changes in the revised manuscript.
Reviewer #3
The authors present a demonstration of concept for a high sensitivity viral protein detection, with the development of the accompanying set of accessories to be used a high throughput field screen test.
Comment 1: This original approach is conceptually intriguing but bears practical challenges in field implementation. The limiting factor of this approach seems to be the preparation of cell culture with the membrane inserted antibody, as the culture to be used for the diagnostic test will need to be prepared de novo for each batch, thus presuming tissue culture lab, reasonably adjacent to the test site. The use of live transiently modified cells also raises several questions as to the feasibility of the high throughput test that should be addressed in the manuscripts: what would be the setup and practical scale for producing these cells, how often the cells need to be regenerated (practically), how long they can stay in culture to preserve they sensing capabilities, can they be frozen, how would time after the modification affects the sensitivity of the cells? This should be discussed in the context of a precedent (or the lack of it).
Response: Indeed, cell-based biosensors present unique challenges for mass-scale, regular on-site applications. The most critical issue is reduced cell viability. Various approaches have been proposed to overcome this limitation, including the use of microfluidic/organ-on-chip circuits in biosensing platforms or specific cell types (e.g. fish gills). However, these approaches have not been yet proven to be suitable for practical, cost-efficient operation of cell-based biosensors to routine, mass-screening applications.
Based on our past experience with commercial applications of membrane-engineered cells and according to the results of the present study, we know that cell viability and biosensor response remain unaltered at least three days after electroinsertion. Cells prepared on day 1 could be shipped and used on-site during the next two days. As it stands, this set up allows for using the biosensor by relatively locally situated end-users, preferably availing over controlled environments (e.g. diagnostic laboratories). We have calculated the actual logistics of the capacity of a cell culture laboratory to produce enough biorecognition elements to meet the local (e.g. within a city or small region) screening requirements. At least 45x106 membrane-engineered cells could be manufactured by a skilled technician on a daily basis, corresponding to more than 100 tests. We have previously reported [42] that the immobilization of membrane-engineered cells in calcium alginate gels and their consequent storage in medium supplemented or not with 20% (v/v) foetal calf serum enabled the preservation of at least 90% of viable cells four weeks after electroinsertion. Cell proliferation could be controlled by adjusting the concentration of foetal calf serum. This has allowed the practical shipment and use of immobilized membrane-engineered cells by remote end-users in a number of applications. Cell immobilization in gel has not been used in the present report; we definitely plan to investigate this possibility in the next round of the biosensor development and optimization.
Action taken: The following statement was added in the end of the Discussion of the revised manuscript (lines 322-343):
Cell-based biosensors present unique challenges for mass-scale, regular on-site applications. The most critical issue is reduced cell viability. Various approaches have been proposed to overcome this limitation, including the use of microfluidic/organ-on-chip circuits in biosensing platforms or specific cell types (e.g. fish gills). However, these approaches have not been yet proven to be suitable for practical, cost-efficient operation of cell-based biosensors to routine, mass-screening applications.
Based on our past experience with commercial applications of membrane-engineered cells and according to the results of the present study, we know that cell viability and biosensor response remain unaltered at least three days after electroinsertion. Cells prepared on day 1 could be shipped and used on-site during the next two days. As it stands, this set up allows for using the biosensor by relatively locally situated end-users, preferably availing over controlled environments (e.g. diagnostic laboratories). We have calculated the actual logistics of the capacity of a cell culture laboratory to produce enough biorecognition elements to meet the local (e.g. within a city or small region) screening requirements. At least 45x106 membrane-engineered cells could be manufactured by a skilled technician on a daily basis, corresponding to more than 100 tests (including negative and positive controls). We have previously reported [42] that the immobilization of membrane-engineered cells in calcium alginate gels and their consequent storage in medium supplemented or not with 20% (v/v) foetal calf serum enabled the preservation of at least 90% of viable cells four weeks after electroinsertion. Cell proliferation could be controlled by adjusting the concentration of foetal calf serum. This has allowed the practical shipment and use of immobilized membrane-engineered cells by remote end-users in a number of applications. Cell immobilization in gel has not been used in the present report; we definitely plan to investigate this possibility in the next round of the biosensor development and optimization.
Additional reference:
[42] Katsanakis N,; Katsivelis, A; Kintzios, S. Immobilization of Electroporated Cells for Fabrication of Cellular Biosensors: Physiological Effects of the Shape of Calcium Alginate Matrices and Foetal Calf Serum. Sensors 2009, 9, 378-385. DOI: 10.3390/s90100378
Comment 2: Establishing the variability between iterations of membrane engineering step is critical as this step needs to be repeated for every batch. This variability of the transient cell modification will also emanate from dilution of the probe as cells divide.
Response: It is entirely accurate that the membrane engineering process must be repeated for each batch of measurements. The protocol applied for membrane engineering has been well established and tested in several applications during the last 15 years. In the present study, we used membrane-engineered cells the following day after the electroinsertion process. The overwhelming majority of these cells were viable (see also response to Comment #2, Reviewer #2). However, it should be expected that cell viability and membrane-engineering efficiency will differ, even slightly, between different batches of engineered cells, not at least due to the different passages of cultured cells. For this reason, we included various sets of controls (non-electoporated cells, electroporated but non-engineered cells) in each individual measurement within every batch of cells.
Action taken: The following statement was added in the end of the Discussion section of the revised manuscript (lines 316-321):
In spite of the fact that membrane engineering has been methodologically standardized and tested in several applications during the last 15 years, it should be expected that cell viability and membrane-engineering efficiency will differ, even slightly, between different batches of engineered cells, not at least due to the different passages of cultured cells. For this reason, various sets of controls (non-electoporated cells, electroporated but non-engineered cells) must be included in each individual measurement within every batch of cells.
Comment 3: Fig. 2 Vero10 is significantly higher than Vero5, as plateau was not reached, would Vero15 be better?
Response: We opted not to test for higher anti-S1 antibody to be electroinserted in membrane-engineered Vero cells due to practical reasons; simply put, we considered that using antibody concentrations higher than 10 μg/mL would render the assay more significantly more costly than desired for eventual scale-up application.
Comment 4: Though qualitatively the response of the modified cells to the antigen exposure is clear, the estimation of variance and reproducibility are key for a tool aiming at mass diagnostics. Therefore, statistics calculations should be presented in greater detail, including the test for significance and the variance estimate (error bar description, what is the nature and number of repeats and statistic parameter they represent, in the legends).
Response: The data presented in the submitted manuscript resulted both from replications of each experiment as well as experiments conducted on different dates. Additional experiments were carried out following the submission of the manuscript in order to possibly extend the range of linear response. In this respect, we were able to determine a linear range of response beyond 100 pg/mL and up to 1 μg/mL. In addition, the established limit of detection was confirmed, the biosensor’s response at the LOD being significantly greater than 3 times the standard deviation of the response against control (see also response to Comment #2, Reviewer #1).
Action taken:
- The following was added in 2.5. Data analysis and experimental design (line 171-173 of the revised manuscript):
…, i.e. the response of each sample x biorecognition element combination was measured eight times by independent assays, while each experimental set was repeated at three different dates (n=24).
- The number of replications (n= 24) was also added in the end of the Figure legends.
- The range of linear response was determined to be 10 fg/mL – 1 μg/mL. This was respectively reported in subsection 3.2 of the revised manuscript (lines 208-210).
- The following was added in the end of 3.2.:
…. which was significantly greater than 3 x standard deviation of the response against control.
This was additionally indicated on the revised Figure 4 (formerly Figure 3).

Round 2
Reviewer 1 Report
This revised manuscript can be accepted.
Reviewer 2 Report
The authors have responded to all of my concerns in full, the ms can be accepted now.
Reviewer 3 Report
The authors addressed the comments